# Microemulsion–Assisted Synthesis of Ag$_2$CrO$_4$@MIL–125(Ti)–NH$_2$ Z–Scheme Heterojunction for Visible–Light Photocatalytic Inactivation of Bacteria

Haoyu Yuan [1,2], Chao Zhang [1,2], Wenjing Chen [2], Yuzhou Xia [2], Lu Chen [2], Renkun Huang [2], Ruiru Si [3] and Ruowen Liang [1,2,*]

1    State Key Laboratory of Photocatalysis on Energy and Environment, Fuzhou University, Fuzhou 350002, China; yuanhaoyu4@163.com (H.Y.)

2    Fujian Province University Key Laboratory of Green Energy and Environment Catalysis, Ningde Normal University, Ningde 352100, China

3    Fujian Key Laboratory of Agro–Products Quality and Safety, Fujian Academy of Agricultural Sciences, Fuzhou 350003, China

*    Correspondence: rwliang@ndnu.edu.cn; Tel.: +86–591–8377–9362

**Abstract:** Metal–organic frameworks (MOFs) are new porous materials composed of metal centers and organic ligand bridges, which received great attention in the field of photocatalysis. In this work, Ag$_2$CrO$_4$@MIL–125(Ti)–NH$_2$ (denoted as AgCr@M125) Z–scheme heterojunctions were synthesized via a simple microemulsion method, by which highly dispersed nano–sized Ag$_2$CrO$_4$ can be anchored uniformly on the surfaces of porous MIL–125(Ti)–NH$_2$ (denoted as M125). Compared with pure M125 and Ag$_2$CrO$_4$, the as–prepared AgCr@M125 hybrids show significant photocatalytic efficiency against inactivated *Staphylococcus aureus* (*S. aureus*), reaching over 97% inactivation of the bacteria after 15 min of visible light irradiation. Notably, the photocatalytic activity of the obtained 20%AgCr@M125 is about 1.75 times higher than that of AgCr–M125, which was prepared via a traditional precipitation method. The enhanced photocatalytic antibacterial activity of the AgCr@M125 photocatalytic system is strongly ascribed to a direct Z–scheme mechanism, which can be carefully discussed based on energy band positions and time–dependent electron spin response (ESR) experiments. Our work highlights a simple way to enhance the antibacterial effect by coupling with Ag$_2$CrO$_4$ and M125 via a microemulsion–assisted strategy and affords an ideal example for developing MOFs–based Z–scheme photocatalysts with excellent photoactivity.

**Keywords:** MIL–125(Ti)–NH$_2$; Ag$_2$CrO$_4$; *S. aureus*; Z–scheme; microemulsion method





## 1. Introduction

With the rapid development of industrialization and the increase in population density year by year, bacterial pollution has become a major threat to human health [1]. Traditional disinfection methods, such as chemical disinfection, thermal radiation, and physical disinfection are difficult to apply widely because they are environmentally unfriendly and expensive. Photocatalysis is considered a prospective and ecofriendly method for the elimination of harmful pollutants from wastewater, which has attracted widespread attention [2]. In the past decades, various materials, including inorganic metal oxides, metal sulfides, and organic semiconductors have been developed as photocatalysts. However, the available photocatalysts are still far from being efficient enough for actual applications. Therefore, exploring more high–performance photocatalysts to inactivate bacteria remains a great commercialization priority for photocatalysis [1–3].

Metal–organic frameworks (MOFs) are an emerging class of crystalline porous materials, formed by inorganic nodes and multifunctional organic linkers. Due to their fascinating characteristics, MOFs have been demonstrated to be outstanding photocatalysts. Particularly, Ti–based MOFs have raised much attention in the field of photocatalysis [4]. For

example, MIL–125(Ti)–NH$_2$ (simply labeled as M125) has a narrow band gap (2.6~2.8 eV) and exhibits significant photo availability to initiate chemical photocatalytic reactions. Fu et al. first investigated MIL–125(Ti)–NH$_2$ to facilitate the photoreduction of carbon dioxide under visible light illumination by using triethanolamine as an electron donor [4]. However, the shortage of active sites and the weak separation resolution of photoinduced vectors decrease the photocatalytic performance of neat MOFs. To further enhance their photoactivity, many strategies have been proposed, such as ligand functionalization, dye sensitization, carbon material decoration, and metal decoration [5]. Incorporating the second semiconductor to construct binary MOFs–based heterostructures [5,6] is supposed to be a superior method to improve the optical properties and boost the charge separation efficiency of MOFs photocatalysts. Z–scheme heterojunction structure is an advanced charge transport mechanism, which retains the valence band (VB) and conduction band (CB) [6] at their highest potential levels in the composite. Interestingly, coupling two semiconductors capable of matching band–edge positions to construct an artificial Z–scheme photocatalytic system is an efficient and controllable method for improving photocatalytic performance that promotes spatial separation of photogenerated carriers while retaining excellent redox capabilities [7].

Choosing the stable and band structure and matching the second component semiconductor are crucial for achieving hierarchically Z–scheme heterojunction. Ag$_2$CrO$_4$ has caught the attention of researchers in recent decades due to its narrow band gap (~1.75 eV) and high photocatalytic activity [8]. In addition, its unique crystalline structure of large O–Ag–O angle and long O–Ag chemical bond makes the charge transfer easier. It is well known that structure determines performance, and the activity of photocatalysts is usually limited by their structure and morphology, which is largely controlled by the synthesis method [9]. When using traditional precipitation methods, it is easier to obtain relatively large size particles, which tend to agglomerate and exhibit uneven size distribution. Microemulsions, as an anisotropic and thermodynamically robust system, have been extensively used to make monodisperse, nanoscale particles with controlled morphology, which can shorten the diffusion process for photogenerated excitons and improve the utilization of light [9,10]. Although various MOFs–based Z–scheme heterojunction photocatalysts, such as FeO$_3$/g–C$_3$N$_4$ [11], C@HPW/CdS [12], and WO$_3$/BiOBr [13], have been reported for the past few years, there is rare research on the Ag$_2$CrO$_4$@MOFs composites and their photocatalytic disinfection performance [14–16].

Herein, we constructed Ag$_2$CrO$_4$@MIL–125(Ti)–NH$_2$ (simply labeled as AgCr@M125) composites in a convenient microemulsion–assisted way for the first time. The photocatalytic antibacterial (taking *S. aureus* as an example) was used as a probe reaction to explore the photocatalytic performance of AgCr@M125. The experimental results showed that the activity of the 20%AgCr@M125 was significantly increased, which could sterilize *S. aureus* within 15 min. The experimental results show that the reaction mechanism follows the Z–scheme model, which enables the material to have efficient photogenerated carrier separation and strong redox ability. This novel AgCr@M125 hybrids not only provide new insights for environmental remediation but also highlights a promising strategy to design more efficient Z–scheme photocatalysts.

## 2. Results and Discussion

### 2.1. Characterisations

As shown in Figure 1a, all the characteristic peaks of M125 are well consistent with previous reports [4,17]. Pristine Ag$_2$CrO$_4$ shows a high crystallinity and the characteristic peaks originated from its orthorhombic structure (JCPDS No. 26-0952). From the enlarged powder X–ray diffraction (XRD) pattern (Figure 1b), the XRD patterns of AgCr@M125 composites are similar to M125. At the same time, the crystalline peaks of (220), (031), (211), and (002) belonging to Ag$_2$CrO$_4$ can be detected, which can also prove the coexistence of Ag$_2$CrO$_4$ and M125 in these composites [18]. The intensity of the peaks corresponding to Ag$_2$CrO$_4$ in AgCr@M125 is quite faint, and it can be attributed to the low crystallinity or

small size of $Ag_2CrO_4$ particles. On the whole, the peak intensity of $Ag_2CrO_4$ gradually increases with the increase of $Ag_2CrO_4$ content [19].

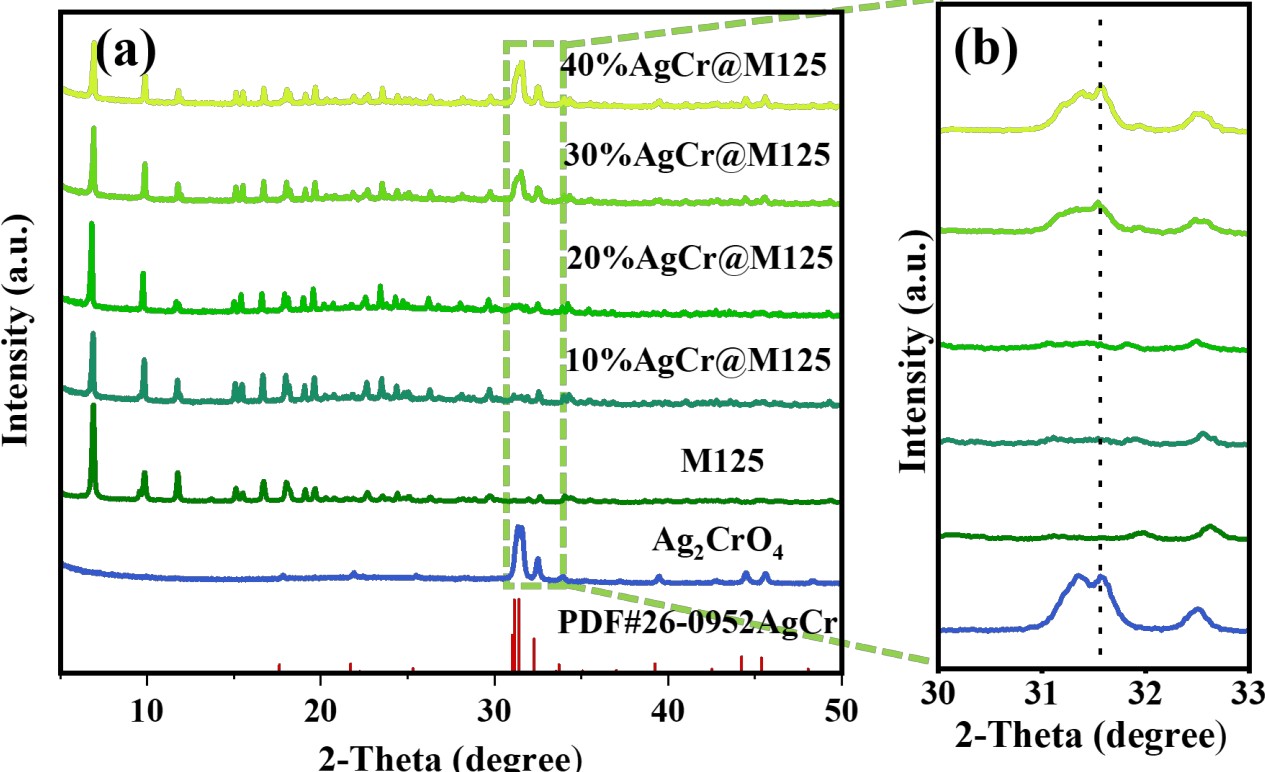

**Figure 1.** XRD patterns of (**a**) $Ag_2CrO_4$, M125, and AgCr@M125 composites; (**b**) enlarged XRD patterns in the range of 30–33°.

The morphologies and microstructures of the samples were analyzed by scanning electron microscopy (SEM) and the results are illustrated in Figure 2. As shown in Figure 2a, the pure M125 shows a smooth pie–like structure with a diameter of about 500 nm (Figure 2a) [20]. The $Ag_2CrO_4$ particles prepared by the precipitation method show an irregular octahedral structure with a grain size of around 200 nm. In the case of the 20%AgCr@M125, the smooth surfaces of the M125 are roughened after coating with the $Ag_2CrO_4$, indicating that the tiny $Ag_2CrO_4$ particles uniformly distribute on the surface of M125. On the contrary, for the 20%AgCr–M125, the $Ag_2CrO_4$ particles decorating the surface of M125 tend to aggregate, indicating the superiority of our microemulsion–assisted method. The microstructure of AgCr@M125 composites was further investigated and analyzed using transmission electron microscopy (TEM) and high–resolution TEM (HRTEM). As shown in Figure 2e,f, $Ag_2CrO_4$ nanoparticles with a diameter of about 10 nm are uniformly dispersed on the surface of M125 [21]. A set of lattice stripes can be seen in Figure 2g with a spacing of about 0.23 nm, corresponding to the (112) lattice spacing of the $Ag_2CrO_4$ orthorhombic phase [22,23]. It is of interest that the microemulsion–assisted growth of AgCr@M125 possesses a more uniform size compared with the conventional precipitation method. As for the 20%AgCr–M125, the particle sizes of $Ag_2CrO_4$ particles are about 20~100 nm and become more severely agglomerated, which also coincides with the SEM results. In order to obtain a more visual confirmation of the atomic distribution, the TEM–mapping was performed. As illustrated in Figure 2m,n, AgCr@M125 prepared by the microemulsion method presents a symmetrical Ag elements distribution and uniform grain size, indicating a uniform distribution of Ag particles. On the other hand, for the AgCr–M125, Ag elements are heavily agglomerated and the grain size varies greatly on the surface of AgCr–M125.

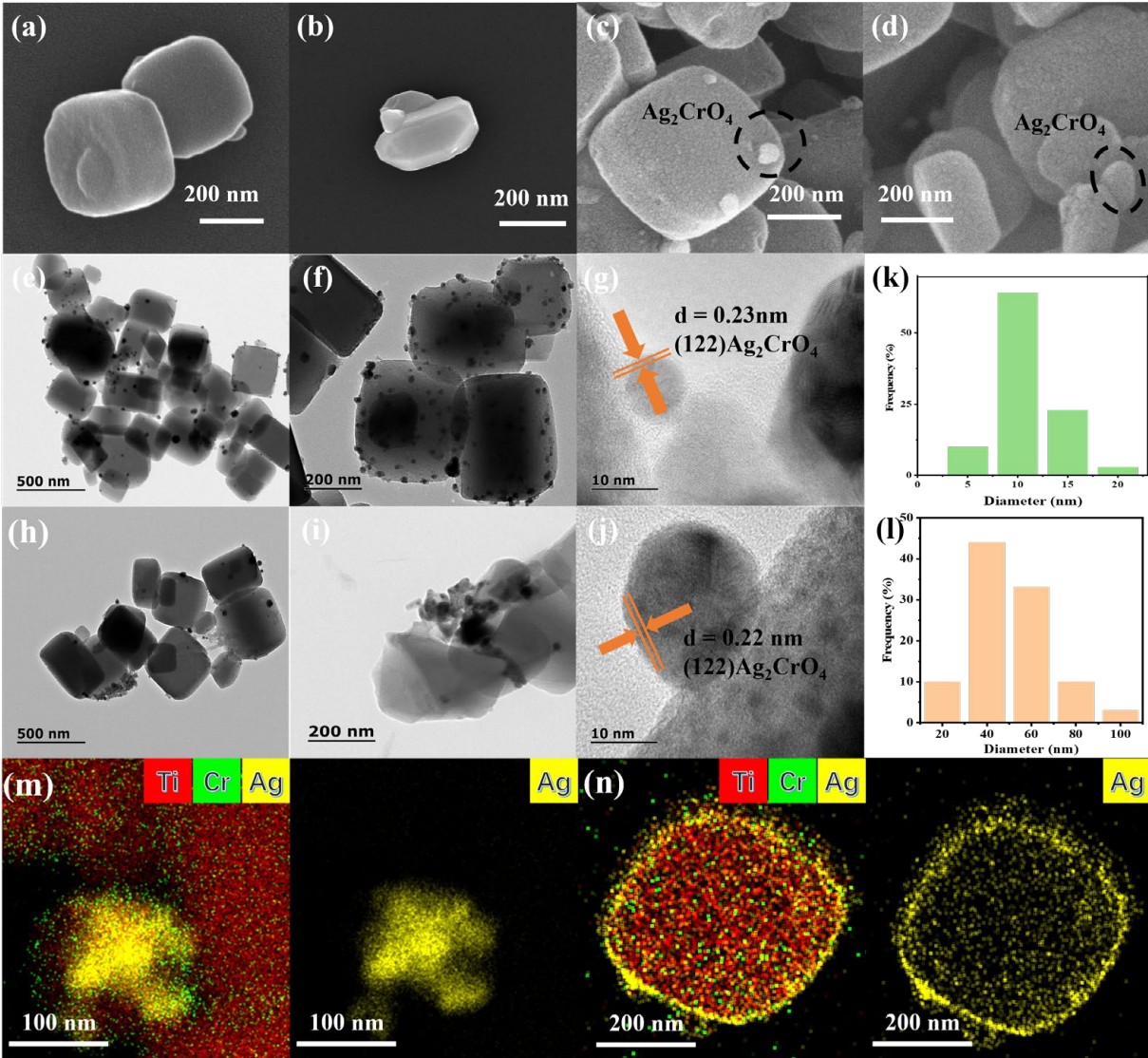

**Figure 2.** SEM images of (**a**) M125, (**b**) Ag$_2$CrO$_4$, (**c**) AgCr@M125, and (**d**) AgCr–M125; TEM images of (**e**–**g**) AgCr@M125 and (**h**–**j**) AgC–M125; the corresponding size distribution of Ag$_2$CrO$_4$ particles (**k**) AgCr@M125 and (**l**) AgCr–M125; the mapping of (**m**) AgCr–M125 and (**n**) AgCr@M125.

The chemical structure was demonstrated by Fourier transform infrared (FTIR) spectroscopy. Figure 3a shows the FTIR spectra of the samples, where the characteristic peak at 3427 cm$^{-1}$ in the spectrum of M125 is the stretching of the surface hydroxyl group, and the peaks between 1380–1600 and 400–800 cm$^{-1}$ [24] are attributed to the stretching vibrational modes of the carboxylate and O–Ti–O, respectively. The FTIR spectra of AgCr@M125 are similar to that of M125 due to the absence of specific functional groups on the surface of Ag$_2$CrO$_4$, which also proves that the process of preparation using the microemulsion method did not destroy the structure of M125. Brunauer–Emmett–Teller (BET) surface areas of M125, Ag$_2$CrO$_4$, and AgCr@M125 samples were determined by adsorption–desorption measurements, and the results are displayed in Figure 3b. According to the BET classification, it can be observed that both M125 and AgCr@M125 adsorption–desorption belong to type IV isotherm [25], which indicates the mesoporous character of the samples. After Ag$_2$CrO$_4$ decorating, the specific surface area of AgCr@M125 decreased, illustrating the surface Ag$_2$CrO$_4$ loading partially blocked the original channel in M125.

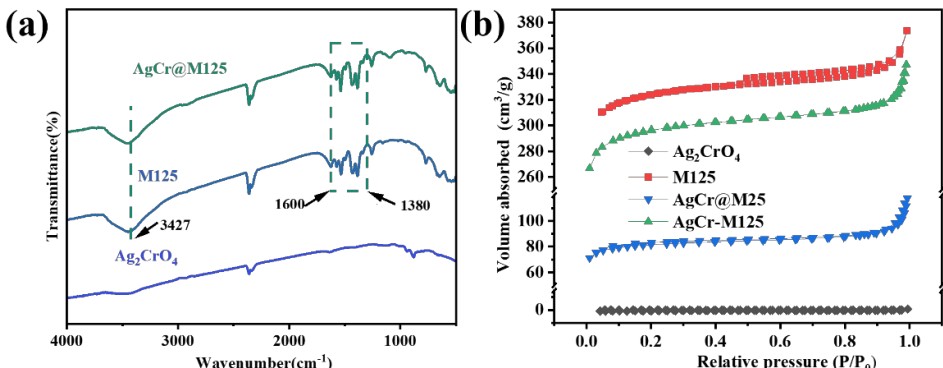

**Figure 3.** (**a**) FT–IR spectra of $Ag_2CrO_4$, M125, and AgCr@M125; (**b**) $N_2$ adsorption–desorption isotherms for different catalysts.

The surface chemical compositions of AgCr@M125 composites were analyzed by X–ray photoelectron spectroscopy (XPS). The survey spectrum of AgCr@M125 shows that the synthesized composites contain five elements (Ti, C, O, N, and Ag) (Figure 4a). From Figure 4b, two characteristic peaks, 402.08 eV and 399.28 eV, are isolated from the high–resolution N 1s spectra. Compared to pure M125, two main peaks of Ti $2p_{1/2}$ and Ti $2p_{3/2}$ in the AgCr@M125 spectrum (464.48 eV and 458.68 eV) are pushed toward low binding energy (Figure 4c) [26]. In Figure 4d, a doublet Cr 2p peak of pure Ag2CrO4 at 576.08 eV (Cr $2p_{3/2}$) and 585.89 eV (Cr $2p_{1/2}$) is attributed to $Cr^{6+}$, which is positively shifted for the AgCr@M125. Moreover, two peaks in the Ag 3d spectra of pure $Ag_2CrO_4$ can be distinguished at 367.5 and 373.5 eV, corresponding to the Ag $3d_{5/2}$ and Ag $3d_{3/2}$ of $Ag^+$ (Figure 4e), respectively [27]. In the Ag 3d spectrum of the AgCr@M125 sample, these peaks are also positively shifted to 368.28 and 374.28 eV, respectively. A similar phenomenon has also been reported in Ag/MIL–125, $Ag_2CrO_4$/SCN, and $Ag_2CrO_4$/g–$C_3N_4$ composites [28–30]. Such transfer in binding energy suggests a robust interaction between $Ag_2CrO_4$ and M125 in AgCr@M125 rather than a simple physical mixing.

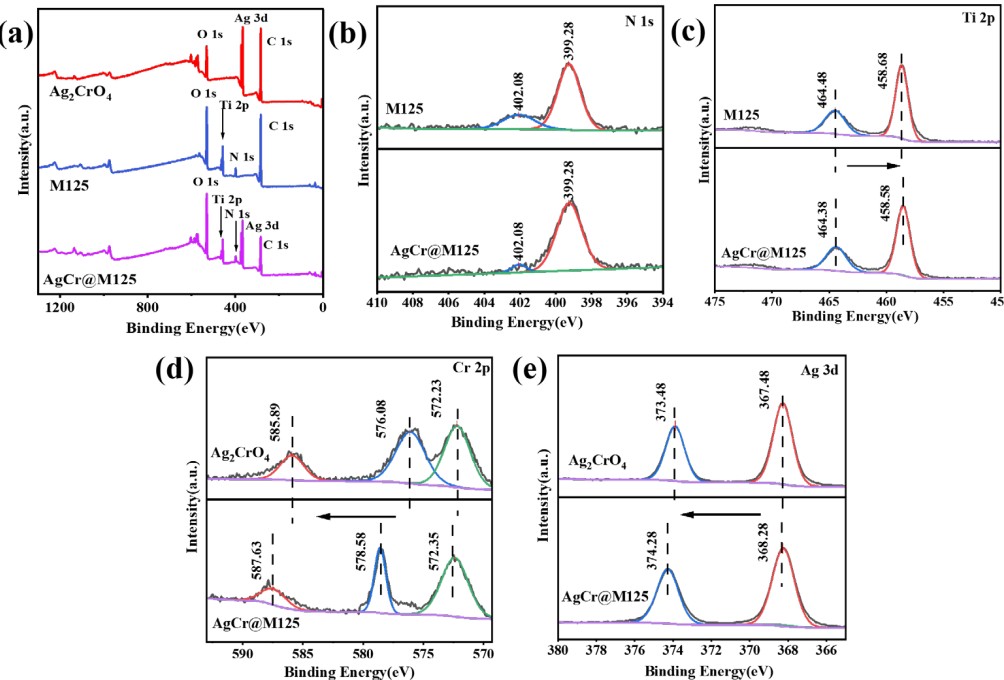

**Figure 4.** XPS spectra of M125, $Ag_2CrO_4$, and AgCr@M125. (**a**) Survey scan; (**b**) N 1s high–resolution scan of (**c**) Ti 2p, (**d**) Cr 2p, and (**e**) Ag 3d.

## 2.2. Energy Band Structure

As can be seen in Figure 5a, the optical absorption edges of the original M125 and Ag$_2$CrO$_4$ are around 450 and 700 nm, respectively [28,31]. The light absorption capacity of the AgCr@M125 composites is higher than that of pure M125 and gradually increases with the increase of Ag$_2$CrO$_4$ content, meaning increased responsiveness to visible light. However, there is no significant change in the absorption edge of all composites, indicating Ag$_2$CrO$_4$ is not incorporated into the lattice of M125 and does not affect the structure of M125 itself. In addition, the band gap energies of all samples can be obtained by the following equation:

$$(\alpha h\nu)^n = A(h\nu - E_g) \tag{1}$$

where *n* is judged by the type of optical jump of the semiconductor (indirect (*n* = 1/2) or direct (*n* = 2)). According to Equation (1), the band gap energies of M125 and Ag$_2$CrO$_4$ are 2.75 eV and 1.87 eV, respectively (Figure 5b).

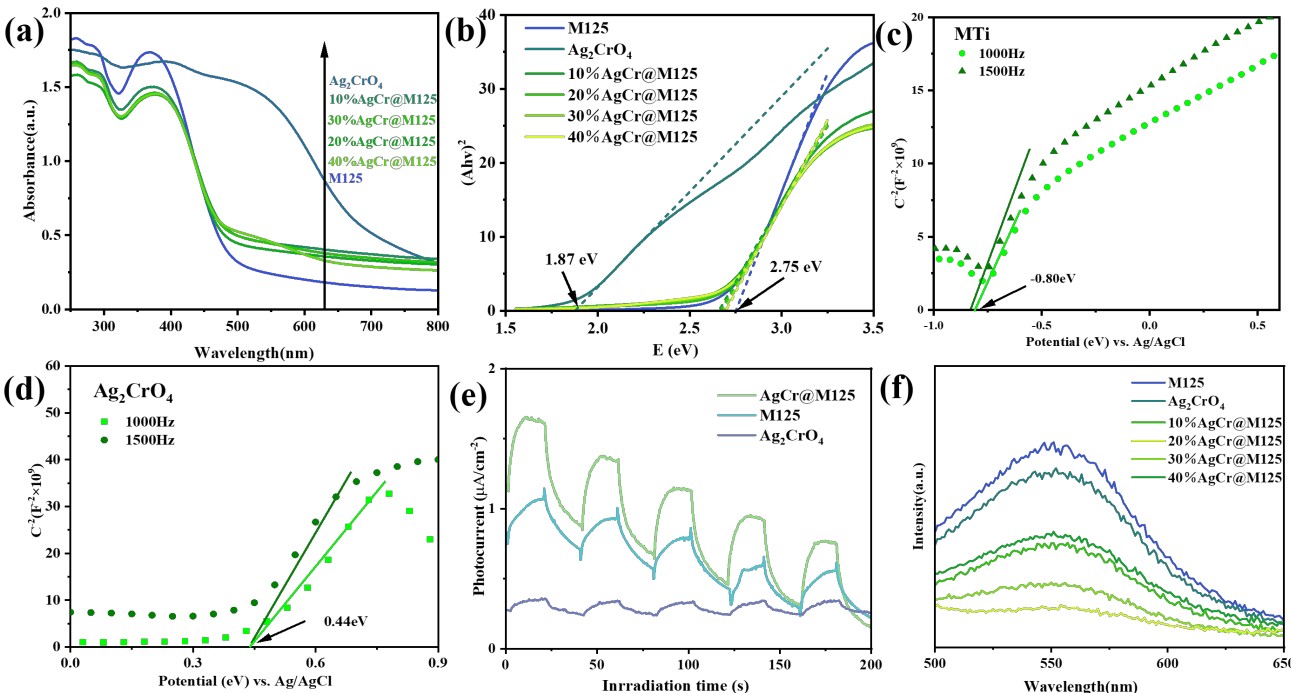

**Figure 5.** (**a**) UV–vis DRS spectra; (**b**) bandgap energy calculation of the as–obtained samples; Mott–Schottky plots of (**c**) Ag$_2$CrO$_4$, (**d**) M125, (**e**) transient photocurrent, and (**f**) photoluminescence spectra of Ag$_2$CrO$_4$, M125 and 20%AgCr@M125.

To calculate the flat band potential (E$_{fb}$) of the original samples, Mott–Schottky (MS) plots were employed. The fact that the C$^{-2}$–E plot has a positive slope indicates that M125 and Ag$_2$CrO$_4$ both exhibit the typical behavior of *n*–type semiconductors. For M125 and Ag$_2$CrO$_4$, the equivalent E$_{fb}$, or conduction band potential (E$_{CB}$), the values are −0.96 eV and +0.36 eV for Ag/AgCl at pH = 6.8, respectively. They are equivalent to −0.77 eV and −0.84 eV versus the typical hydrogen electrode (pH = 6.8). According to Equation (2):

$$E_{CB} = E_{VB} - E_g \tag{2}$$

The respective valence band potential (E$_{VB}$) for M125 and Ag$_2$CrO$_4$ are calculated as −0.6 eV and 0.64 eV, respectively. The photogenerated carrier separation capability of AgCr@M125 was evaluated by the photocurrent response [29–32]. As shown in Figure 5e, the photocurrent intensity of AgCr@M125 is significantly higher than that of the original M125 and Ag$_2$CrO$_4$, which means that this composite can separate and transfer photogenerated carriers more efficiently. As the separation of photogenerated electron–hole

pairs is another critical factor affecting photocatalytic activity, the photogenerated carrier's separation efficiency is demonstrated by photoluminescence spectroscopy. As can be seen from Figure 5f, the peak intensity of 20%AgCr@M125 is lower than those of the pure substances, implying the lowest recombination rate of photogenerated electron–hole pairs for 20%AgCr@M125.

### 2.3. Antibacterial Activity

*S. aureus* photocatalytic disinfection was chosen as a model response to confirm the benefits of the AgCr@M125 hybrids [1,33]. The self–disinfection of *S. aureus* is essentially nonexistent in the absence of a photocatalyst, as shown in Figure 6a. After 15 min of visible light irradiation, the percentages of bacterial survival for individual M125 and $Ag_2CrO_4$ are 44.8% and 51.5%, respectively. Interestingly, all the AgCr@M125 hybrids show enhanced photocatalytic performances compared with the pristine M125, $Ag_2CrO_4$, and 20%AgCr–M125 under identical testing conditions. Only 55.6% of the bacteria can be destroyed by the 20%AgCr–M125, which is prepared by the precipitation method because $Ag_2CrO_4$ aggregates on the surface of M125 and due to the quick recombination of photogenerated electrons and holes. As shown in Figure 6c, the 20%AgCr@M125 sample shows the best antibacterial effect with <3% bacterial survival. Interestingly, when the content of $Ag_2CrO_4$ exceeds 20 wt%, it decreases the photocatalytic performance. As shown in Figure 6c, the disinfection efficiency against *S. aureus* decreased from 97.1% to 67.2%. This result is reasonable because, as the $Ag_2CrO_4$ content increases, it makes the $Ag_2CrO_4$ particles on the surface of M125 start to aggregate, while the excess $Ag_2CrO_4$ particles also shield the active sites on M125, which is consistent with the previous reports [34]. In comparison with other photocatalytic antibacterial studies, the 20%AgCr@M125 exhibits better or comparable activity for photocatalytic antibacterial activity under visible light irradiation (Table 1).

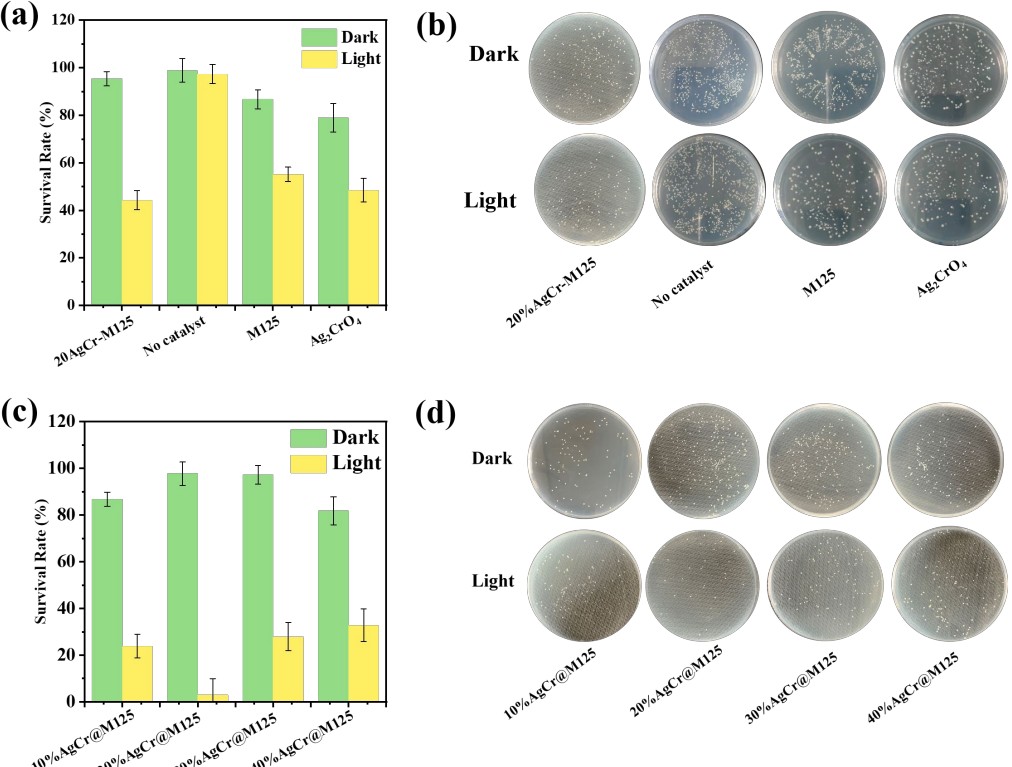

**Figure 6.** Antibacterial effect of AgCr@M125. (**a**,**c**): the survival rate of *S. aureus* after using different photocatalysts; (**b**,**d**): the colony plots of *S. aureus* after treatment with different photocatalysts.

**Table 1.** Comparison of the photocatalytic antibacterial (*S. aureus* activities of 20%AgCr@M125 composites and other reported catalysts.

| Photocatalyst | Dosage (g/L) | Irradiation Time (min) | Light Wavelength | Efficiency (%) | Ref. |
|---|---|---|---|---|---|
| 20%AgCr@M125 | 0.2 | 15 | 300 W ($\lambda > 420$ nm) | 97.7 | This work |
| $TiO_2$–$NH_2$@Au NC | 1.2 | 60 | 300 W ($\lambda > 400$ nm) | 99.9 | [35] |
| g–$C_3N_4$–V–$TiO_2$ | 0.5 | 60 | 500Wvis | 99.5 | [36] |
| $Ag_2S$/NCs | 0.1 | 60 | NIR irradiation (808 nm) | 97.3 | [37] |
| $ZnCl_2$/$TiO_2$, | 4.0 | 120 | 270 Wvis | 90.0 | [38] |
| CQDs/g–$C_3N_4$ | 1.0 | 220 | 300 W ($\lambda > 400$ nm) | 99.9 | [39] |
| $BiVO_4$–C300 | 1.0 | 120 | 300 Wvis | 72.8 | [40] |
| Mg/ZnO | 2.0 | 180 | 300 Wvis | 90.0 | [41] |
| g–$C_3N_4$–AgBr | 0.1 | 150 | 300 Wvis | 99.9 | [42] |

Reusability and stability are important properties to examine the practicability of photocatalysts [43,44]. As shown in Figure 7a, the 20%AgCr@M125 exhibit a stable photocatalytic performance with no obvious activity of decaying after four cycles of photocatalytic reactions. The powder XRD patterns and TEM images confirm that the crystal structure and morphology of 20%AgCr@M125 are not destroyed after cycling experiments (Figure 7b,c), meaning that AgCr@M125 has good stability under visible light irradiation.

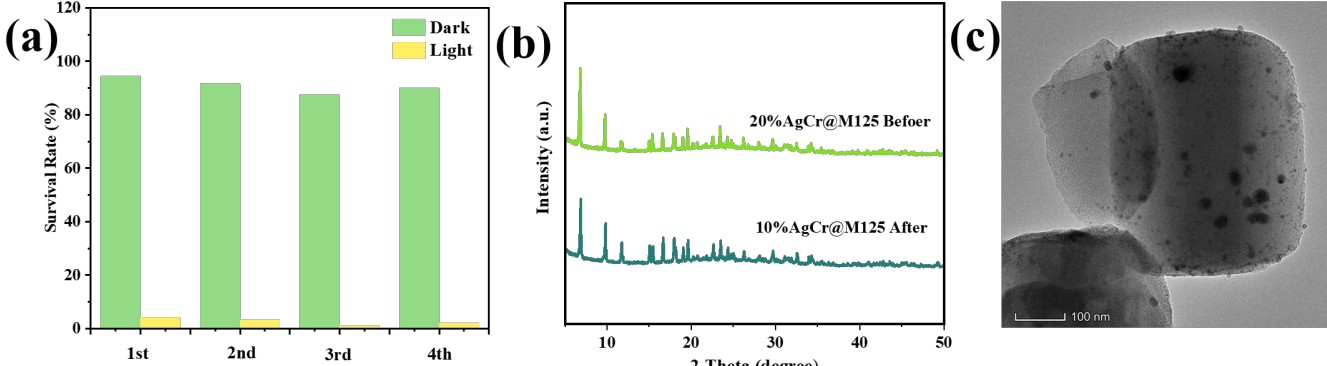

**Figure 7.** (**a**) Recyclability of 20%AgCr@M125 for the photocatalytic disinfection of *S. aureus* under visible light irradiation; (**b**) XRD patterns; (**c**) TEM image of 20% AgCr@M125 after the reaction.

### 2.4. Possible Photocatalytic Mechanism

To confirm the direct Z–scheme heterojunction mechanism, the electron spin resonance (ESR) technique was carried out under visible light. The generation of $h^+$ could be detected by 2,2,6,6–tetramethyl piperidinyl–1–aryloxy (TEMPO) because its radicals could be oxidized by $h^+$. As shown in Figure 8a, the TEMPO characteristic peak decreases significantly with the extension of the light time, meaning that holes are photogenerated during the disinfection reaction. From Figure 8b,c, the radical signals of DMPO−•OH and DMPO−•$O_2^-$ are not detected under dark conditions but appear when exposed to light. According to these findings, the major species involved in the photocatalytic reaction are $h^+$, •$O_2^-$, and •OH.

By a simple comparison of the $H_2O$/•OH and $O_2$/•$O_2^-$ redox potentials versus the CB and VB potentials of M125 and $Ag_2CrO_4$. It can be found that •OH radicals cannot form over pure M125, but can form over pure $Ag_2CrO_4$, as evidenced by the fact that the VB potential of $Ag_2CrO_4$ (2.51 eV vs. NHE) is more positive than that of $E_{(•OH/H2O)}$ (2.27 eV vs. NHE). On the contrary, •$O_2^-$ radicals cannot be generated over $Ag_2CrO_4$, but can be generated at the CB potential of M125 (−0.60 eV for NHE), which is more negative than $E_{(O2/O2•−)}$ (−0.33 eV vs. NHE). If the electron–hole pair at the interface of AgCr@M125 follows the conventional type–II heterojunction mechanism, it is impossible to detect such a strong signal of •OH and •$O_2^-$ radicals because the redox capacity of AgCr@M125

composites is impaired by such type–II structure [45–47]. Hence, we deduce that the charge transfer between $Ag_2CrO_4$ and M125 follows a Z–scheme heterojunction [45–50], and it not only promotes the effectiveness of photocatalytic disinfection but also benefits photogenerated carrier separation.

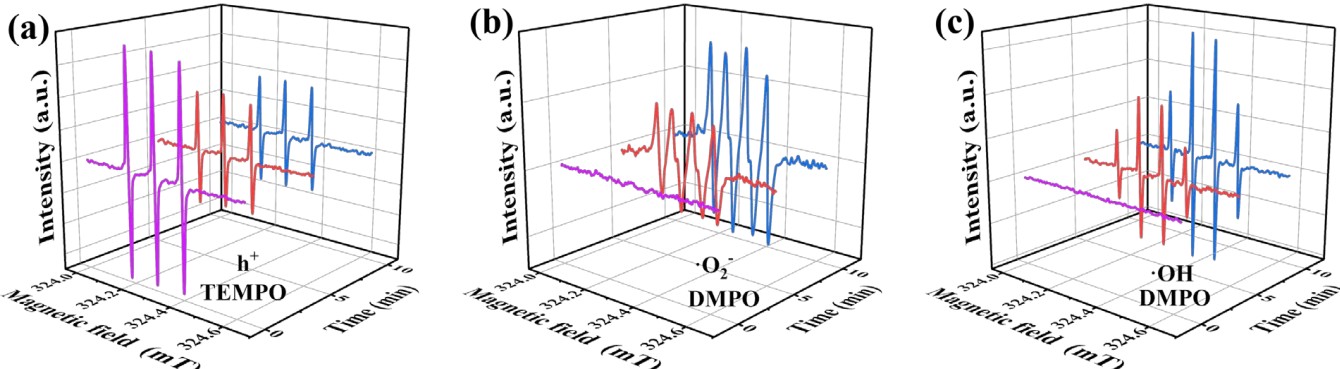

**Figure 8.** ESR spectra upon irradiation of AgCr@M125 heterostructure with different times: (**a**) $h^+$, (**b**) $\bullet O_2^-$, and (**c**) $\bullet OH$.

A possible Z–scheme photocatalytic mechanism is schematically depicted in Figure 9. Under light irradiation, the electrons on the CB of $Ag_2CrO_4$ can easily transfer to the VB of M125 and recombined with the holes of M125 [48–51]. In this way, the photogenerated holes in M125 and the photogenerated electrons in $Ag_2CrO_4$ with poor redox ability can be sacrificed, and the electrons in M125 and the holes in $Ag_2CrO_4$ with strong redox ability remained and acted on the photocatalytic antibacterial. All the relevant reactions were described as follows [52,53]:

$$M125 + hv \rightarrow M125\ (h^+/e^-) \tag{3}$$

$$Ag_2CrO_4 + hv \rightarrow Ag_2CrO_4\ (h^+/e^-) \tag{4}$$

$$e^-\ (M125) + O_2 \rightarrow \bullet O_2^- \tag{5}$$

$$h^+\ (Ag_2CrO_4) + OH^- \rightarrow \bullet OH \tag{6}$$

Photogenerated electrons at CB of M125 react with $O_2$ to produce $\bullet O_2^-$, which is a powerful oxidant in photocatalytic antibacterial. On the other hand, the holes in the VB of $Ag_2CrO_4$ can react with $H_2O$ to generate $\bullet OH$ for favoring *S. aureus* oxidation. Therefore, $\bullet OH$, $\bullet O_2^-$, and the holes stored in the VB of $Ag_2CrO_4$ with strong oxidation ability could effectively kill the *S. aureus*. On the base of the above photocatalytic mechanism discussion, it can be suggested that the Z–scheme heterojunction mechanism can effectively separate photogenerated carriers and has a strong redox ability, thus improving the efficiency of photocatalytic antibacterial.

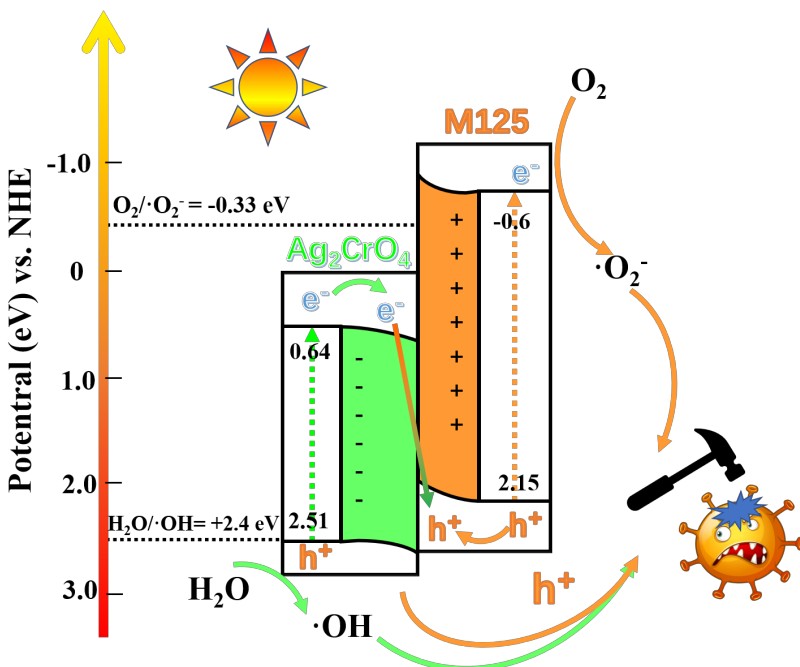

**Figure 9.** Reaction mechanism for photogenerated electron–hole pairs separation and transport over AgCr@M125 composite towards photocatalytic antibacterial.

## 3. Materials and Methods

### 3.1. Materials

All reagents and solvents were provided directly from commercial suppliers without further purification. The 2–amino–terephthalic acid ($H_2ATA$) was obtained from Alfa Aesar Chemical Co., Ltd. (Tianjin, China). Tetra butyl titanate and nutrient agar were obtained from Aladdin Reagent Co., Ltd. (Shanghai, China). Sodium chloride (NaCl), potassium dichromate ($K_2CrO_4$), silver nitrate ($AgNO_3$), cyclohexane, triton X–100, *n*–hexanol, N,N–dimethylformamide (DMF), anhydrous methanol, and anhydrous ethanol were supplied by Sinopharm Chemical Reagent Co., Ltd. (Beijing, China).

### 3.2. Catalyst Preparation

#### 3.2.1. Preparation of M125

M125 (simply labeled as M125) was synthesized according to the literature with some modifications. In brief, 0.5 g of $H_2ATA$, 36 mL of DMF, and 4 mL of anhydrous methanol were added into a 100 mL Teflon–lined autoclave and stirred for 5 min. Subsequently, 1.2 mL of tetra butyl titanate was dripped into the above mixture and then stirred for 30 min. After heating with Teflon at 150 °C for 72 h, the product was left to cool at room temperature naturally. After that, the obtained M125 was collected by centrifuging and washing with DMF and anhydrous methanal several times to completely replace the guest DMF solvent molecule in the pores. Finally, the yellow powder sample was placed in a 70 °C oven for 12 h.

#### 3.2.2. Preparation of AgCr@M125

A total of 16 mL of cyclohexane, 5.2 mL of triton X–100, and 3 mL of hexanol were mixed well at room temperature, followed by the slow addition of 100 mg of M125 and sonication for 30 min. After that, an amount of aqueous $K_2CrO_4$ (0.12 mol/L) was added drop by drop to the suspension to achieve a W/O reversed colloidal system, followed by rapid stirring for 1 h, to allow $CrO_4^{2-}$ to be loaded on the surface of M125. Then, the same volume of $AgNO_3$ aqueous solution (0.24 mol/L) was added and continuously stirred for about 24 h. The precipitate was centrifuged and washed with ethanol and deionized water, and after drying at 70 °C for 24 h the final product was obtained. The theoretical mass

ratios of $Ag_2CrO_4$ were 0.1, 0.2, 0.3, and 0.4, and named 10%AgCr@M125, 20%AgCr@M125, 30%AgCr@M125, and 40%AgCr@M125, respectively. The pristine $Ag_2CrO_4$ was synthesized in a similar procedure but without the addition of M125.

### 3.2.3. Preparation of AgCr–M125

For comparison, AgCr–M125 was synthesized via a self–assembly method. Briefly, 100 mg of M125 was added into a known volume of $K_2CrO_4$ solution (0.12 mol/L) and ultrasonication was conducted for 30 min. Subsequently, an isometric $AgNO_3$ solution (0.24 mol/L) was added to this suspension and stirred for 24 h. The precipitate was centrifuged and washed with ethanol and deionized water, and after drying at 70 °C for 24 h the final products were obtained. The theoretical mass ratio of $Ag_2CrO_4$ was 0.2 and named 20%AgCr–M125.

### 3.3. Catalyst Preparation

Powder X–ray diffraction (XRD) was performed on a Bruker AXS D8 Advance X–ray diffractometer. The morphology of the sample was observed under a ZEISS Sigma 300 scanning electron microscope (SEM) and a US Thermo Fisher Talos F200S G2 transmission electron microscopy (TEM). Fourier transform infrared (FT–IR) spectra were recorded on a Thermo Scientific Nicolet iS10 infrared spectrophotometer. X–ray photoelectron spectroscopy (XPS) was observed on a US Thermo Scientific K–Alpha Science spectrometer. A Shimadzu UV–2700 equipment was used to collect UV–vis diffuse reflectance spectra (UV–vis DRS). The Edinburgh FLS1000 was used to collect photoluminescence spectra. On Micromeritics ASAP 2460 characteristics, the specific surface area of the samples was calculated. A JEOL JES–FA200 spectrometer was used to record electron spins (ESR). To capture the radical's signal, 5,5–dimethyl–1–pyrroline–N–oxide (DMPO) and 2,2,6,6–tetramethyl–1–piperidinyl oxy (TEMPO) were used.

Based on a standard three–electrode cell system, electrochemical measurements were carried out, utilizing a Pt plate and Ag/AgCl electrode as the counter and reference electrode, respectively. The electrolyte was a 0.2 M $Na_2SO_4$ solution aqueous (pH = 7). After sonicating in ethanol for 30 min and drying at 353 K, the working electrode was cleaned using fluorine–doped tin oxide (FTO) glass. To create a slurry for the experiment, 5 mg of the sample was sonicated in 0.5 mL of DMF to ensure uniform sample dispersion. To improve adhesion, the pretreatment FTO glass was evenly covered with a coating of slurry and further dried at 393 K. After that, the scotch tape was removed and the uncoated part of the electrode was isolated with epoxy resin, leaving a 0.25 $cm^2$ exposed area to modify the sample. Bias–free photocurrent measurements were taken on a BAS Epsilon workstation, and A CHI0E A5225 workstation was used for Mott–Schottky texts.

### 3.4. Assessment of Antibacterial Activity

The photocatalytic antibacterial effect was evaluated using *S. aureus* as a targeted contaminant. A concentration of $2 \times 10^9$ colony–forming units (CFU) $mL^{-1}$ of *S. aureus* was used for this study. The photocatalyst (10 mg) was dispersed in 40 mL of saline (0.9% NaCl (*w/v*)) and sterilized using an autoclave. A 300 W Xenon lamp (PLS–SXE300D, Beijing Perfect Light Source Technology Co., Ltd. (Beijing, China.)) with a 420 nm filter was used as the light source for the photocatalytic antibacterial experiments, and samples were taken one at a time with time intervals. The diluted solution (100 μL) was spread flat on an agar plate and incubated at 37 °C for 24 h to assess the inactivation performance by plate counting method.

## 4. Conclusions

In summary, AgCr@M125 nanocomposites were successfully synthesized by a facile microemulsion–assisted method. In this architecture, the highly dispersed $Ag_2CrO_4$ with a size of ~10 nm is evenly dispersed on the M125. Compared with the AgCr–M125 prepared by the precipitation method, the AgCr@M125 has more uniform $Ag_2CrO_4$ particle loading

and higher photocatalytic activity. The optimum composite with 20 wt.% $Ag_2CrO_4$ exhibits the best photocatalytic activity, and its *S. aureus* killing rate reaches 97% after 15 min of visible light irradiation. The enhancement in the photoactivity is attributed to the high dispersion of $Ag_2CrO_4$ particles and the Z–scheme heterojunction between $Ag_2CrO_4$ and M125, which provides an efficient transfer pathway for charge carriers while the composites with strong redox ability are endowed. Finally, a possible antimicrobial mechanism is proposed. This work expands the application of MOF materials and provides a good demonstration of the design of novel MOF–based Z–scheme photocatalysts.

**Author Contributions:** Conceptualization, H.Y. and R.L.; methodology, H.Y., R.L. and Y.X.; investigation, H.Y. and C.Z.; resources, H.Y., R.L., L.C. and Y.X.; data curation, H.Y. and W.C.; writing—original draft preparation, H.Y.; writing—review and editing, H.Y., R.L., R.H., R.S., L.C. and Y.X.; visualization, H.Y. and R.L.; supervision, R.L. and R.S.; project administration, R.H. and L.C.; funding acquisition, R.L. All authors have read and agreed to the published version of the manuscript.

**Funding:** This research was funded by the Scientific Research Fund Project of Ningde Normal University, grant number 2021ZDK06.

**Data Availability Statement:** The data presented in the study are available from the corresponding author.

**Conflicts of Interest:** The authors declare no conflict of interest.

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
