# Peer review of "Microemulsion–Assisted Synthesis of Ag2CrO4@MIL–125(Ti)–NH2 Z–Scheme Heterojunction for Visible–Light Photocatalytic Inactivation of Bacteria"

_catalysts, doi:10.3390/catal13050817_

Round 1

Reviewer 1 Report

1. Abstract is too general, the author should revise the abstract with the obtained results and conclusion.

2. The authors should make a comparison with other studies on the photocatalytic inactivation of bacteria.

3. It is better to add the EDS-SEM mapping for more confirmation of the atomic distribution of AgCr-M125 and AgCr@M125.

4. How did the authors measure the size distribution of Ag2CrO4 (Figure 2 (k) and (l)), considering in situ synthesis of Ag2CrO4?

5. The manuscript needs careful editing and grammar corrections.

Author Response

Thank you for your valuable suggestion very much! We have revised the manuscript carefully according to the suggestions. 

Reviewer 2 Report

The submitted manuscript deals with the issue of Microemulsion-assisted synthesis of Ag2CrO4@MIL-125 and the investigation of its antibacterial activity. In the beginning of the work, the authors describe the issue of Metal-organic frameworks (MOFs) and their use in various areas.

The results of the work are clearly described and explained. For the presentation of the results, the authors used a sufficient number of graphs, supplemented by photographs from individual experiments.

The methodology used is adequate for the given scientific field. Modern analytical methods are used in the work, which are thoroughly described.

The authors of the article succeeded in the successful synthesis of AgCr@M125 nanocomposites with higher photocatalytic activity. The possible antimicrobial mechanism was proposed. The work expands the application of MOFs materials in this area.

Based on the study of the manuscript and the quality of its processing, I have no comments and I recommend publishing it in its entirety without further modifications.

Author Response

(The authors gave the same response as above.)

Reviewer 3 Report

This manuscript is properly written and organized but still needs to be revised and improved for publication.

1.      Why did the authors choose the four ratios to prepare AgCr@M125 composites? 

2.      To further verify the photocatalytic antibacterial activity, PL should be provided for all of the composites.

3.      The 20%AgCr@M125 sample showed the highest antibacterial activity, but the bandgap is almost the same as other samples; why?

4. The authors can see the following papers published and cite them to see if they are of any use during revision: Ceramics International 48 (15), 21939–21946, 2022, doi: https://doi.org/10.1016/j.ceramint.2022.04.176; New Journal of Chemistry 46 (19), 8999–9009, 2022; Catalysts 12 (3), 308, 2022.

Author Response

(The authors gave the same response as above.)
